# Bilateral Diffuse Nodular Pulmonary Ossification Mimicking Metastatic Disease in a Patient with Fibrolamellar Hepatocellular Carcinoma

**DOI:** 10.3390/children8030226

**Published:** 2021-03-16

**Authors:** Pattamon Sutthatarn, Cara E. Morin, Jessica Gartrell, Wayne L. Furman, Max R. Langham, Teresa Santiago, Andrew J. Murphy

**Affiliations:** 1Department of Surgery, St. Jude Children’s Research Hospital, Memphis, TN 38105, USA; Pattamon.Sutthatarn@stjude.org (P.S.); mlangham@uthsc.edu (M.R.L.); 2Department of Surgery, Faculty of Medicine, King Chulalongkorn Memorial Hospital, Chulalongkorn University, Bangkok 10330, Thailand; 3Department of Diagnostic Imaging, St. Jude Children’s Research Hospital, Memphis, TN 38105, USA; cara.morin@stjude.org; 4Department of Oncology, St. Jude Children’s Research Hospital, Memphis, TN 38105, USA; jessica.gartrell@stjude.org (J.G.); wayne.furman@stjude.org (W.L.F.); 5Department of Surgery, University of Tennessee Health Science Center, Memphis, TN 38163, USA; 6Department of Pathology, St. Jude Children’s Research Hospital, Memphis, TN 38105, USA; teresa.santiago@stjude.org

**Keywords:** diffuse nodular pulmonary ossification, fibrolamellar hepatocellular carcinoma, pulmonary calcification

## Abstract

Pulmonary ossification (PO) is a rare finding, characterized by mature bone formation in the lung parenchyma. We report a 20-year-old female patient diagnosed with fibrolamellar hepatocellular carcinoma (FL-HCC) and bilateral diffuse nodular PO. The patient presented with a unifocal left liver mass and multiple bilateral pulmonary lesions, which were treated as metastatic disease. The patient received neoadjuvant chemotherapy, followed by left hepatectomy, and bilateral staged thoracotomies for clearance of the pulmonary disease. The histology of the pulmonary nodules demonstrated nodular type PO. We present the history, the course of treatment, imaging, and histologic findings of this rare disease process that could mimic metastatic pulmonary disease.

## 1. Introduction

Pulmonary ossification (PO) is an uncommon benign pulmonary condition characterized by heterotopic bone deposition in the lung parenchyma. It has an incidence of 1.63 per 100 cases at autopsy in patients without background lung disease [1]. This condition was first described in 1856 by Luschka from autopsy findings. PO can be idiopathic or associated with pulmonary or cardiac disease [2]. PO is histologically characterized by evidence of mature bone formation in the pulmonary parenchyma with or without bone marrow elements (fat or hemopoietic cells). There are two histologic subtypes of PO: nodular and dendritic. The nodular form is composed of lamellar deposits of calcified osteoid material without bone marrow elements in the alveolar spaces. The dendritic form is characterized by interstitial branching spicules of mature bone containing marrow elements [1,3]. The nodular form is usually found in patients with chronically elevated pulmonary venous pressure or left-sided heart failure, such as in patients with mitral valve stenosis [3,4,5]. The dendritic form is more often associated with interstitial pulmonary fibrosis, which can be idiopathic or caused by exposure to asbestos, chemotherapeutic agents [6,7], chronic inflammation, or the chronic aspiration of gastric acid [5,8]. 

In this report, we present the case of a 20-year-old female diagnosed with fibrolamellar hepatocellular carcinoma (FL-HCC) with bilateral multiple pulmonary nodular ossifications, initially interpreted as pulmonary metastases, resulting in an intensification of the systemic therapy and bilateral thoracotomies. FL-HCC is a rare liver cancer that is most common in young adults without preexisting liver disease or cirrhosis [9,10]. FL-HCC often presents at advanced stages at diagnosis and is notably refractory to most conventional chemotherapeutic agents. Surgical resection is the mainstay of treatment for FL-HCC. The diagnosis of FL-HCC is confirmed by detecting the *DNAJB1-PRKACA* fusion transcript, which is pathognomonic for this tumor [11,12]. The Children’s Oncology Group trial AHEP1531, as part of the Pediatric Hepatic International Tumor Trial (PHITT), is currently enrolling patients with newly diagnosed HCC (including FL-HCC). For patients with unresectable or metastatic disease at diagnosis, the trial aims to compare chemotherapy response, resectability, and survival of patients who receive PLADO chemotherapy (cisplatin/doxorubicin) + sorafenib versus PLADO + GEMOX (gemcitabine/oxaliplatin) + sorafenib. 

## 2. Case Presentation

A 20-year-old female was referred to our institution to evaluate and treat a left liver mass with possible bilateral pulmonary metastases. She presented with worsening epigastric abdominal pain over three months. She had no prior contributory medical history, including no history of cirrhosis or liver disease. She endorsed a history of recreational cannabis smoking for three years. Her physical examination was unremarkable. Her alpha-fetoprotein (AFP) at diagnosis was 1.8 ng/mL (normal 0.8–7.5 ng/mL). Serum bilirubin, aspartate aminotransferase (AST), alanine aminotransferase (ALT), and alkaline phosphatase (ALP) were all within normal limits. Hepatitis B serologies and hepatitis C antibody were all non-reactive. 

An abdominal computerized tomography (CT) scan with contrast showed a 6.2 × 4.8 cm tumor involving the left liver, including segments II, III, and IVa. MRI of the abdomen with gadoxetate disodium (Eovist^TM^ or Primovist^TM^) showed a unifocal left hepatic lesion with lobular peripheral enhancement, suspicious for FL-HCC (Figure 1).

CT scan of the chest showed numerous small (3–4 mm) bilateral pulmonary nodules (Figure 2). Because she had two or more non-calcified pulmonary nodules, each greater than or equal to 3 mm in diameter, this met criteria for metastatic disease per the PRETEXT 2017 staging system used on the AHEP1531 protocol. Therefore, the pre-treatment extent of the disease (PRETEXT) stage was determined to be II (+M) [13]. Core needle biopsy of the hepatic mass demonstrated a neoplasm composed of thick trabeculae of large polygonal cells separated by dense desmoplastic collagenous stroma (Figure 3). The individual tumor cells had abundant granular eosinophilic cytoplasm and atypical nuclei with conspicuous eosinophilic nucleoli. Foci of ischemic-type tumor necrosis were observed. Overall, the histopathological features were consistent with FL-HCC. The non-neoplastic liver showed mild lymphocytic infiltration but no fibrosis or cirrhosis. Whole-genome, whole-exome, and transcriptome analysis of the patient’s tumor sample identified a *DNAJB1-PRKACA* fusion (pathognomonic in-frame fusion of *DNAJB1* exon 1 to *PRKACA* exon 2), confirming the diagnosis of FL-HCC.

She was enrolled on the AHEP1531 protocol for HCC, arm F (metastatic or unresectable disease at diagnosis). Her echocardiogram at the beginning of treatment demonstrated normal left ventricular systolic function, normal left atrial size, and no evidence of pulmonary hypertension. She was randomized on arm F to receive three cycles of cisplatin/doxorubicin (PLADO) and sorafenib, a port-a-cath was placed, and neoadjuvant treatment was initiated. After three cycles of neoadjuvant chemotherapy and sorafenib, a CT of the abdomen showed the liver mass to be 5.3 × 3.8 cm, consistent with stable disease by RECIST criteria [14]. Her POSTTEXT stage was II (+M). The pulmonary nodules did not change in size or number, but interim punctate calcification was suspected (Figure 2).

She underwent left hepatectomy with resection of segments II, III, and IV of the liver. The left hepatic vein, left portal vein, left hepatic duct, and left hepatic artery were ligated and divided during the procedure. The gallbladder was resected en bloc with the surgical specimen. Segment I was preserved. The microscopic margins of resection were negative. The pathology was consistent with a post-therapy FL-HCC with approximately 38% viable tumor (Figure 3A,B). 

One month after the liver resection, the bilateral pulmonary nodules remained unchanged in size compared to prior CT. The patient then underwent staged right thoracoscopy converted to open right thoracotomy with intraoperative near-infrared (NIR) image-guided resection. The indocyanine green (ICG) was given 24 h prior to surgery. We had to convert to open thoracotomy in order to perform manual palpation to identify the lesions because the lung nodules were not ICG-avid and not visualized under thoracoscopy [15,16]. There were 11 palpable lung nodules on the right side, which were removed during the first staged thoracotomy. Nine of eleven right lung nodules showed PO (Figure 3C,D), and the other two nodules demonstrated benign intraparenchymal lymph nodes. Three weeks later, the patient underwent a second stage left thoracoscopy. The operation was converted to open left thoracotomy again to palpate the lesions since there were no visible lesions and no ICG avid lesions. We found six nodules that were enucleated, and the histologic examination of the specimens revealed lobulated bone nodules without marrow elements within alveolar lung tissue and no inflammatory, no granulomatous, or tumor cells. These findings were consistent with nodular type PO (Figure 3C,D). No neoplastic cells were observed in any of the 17 pulmonary resection specimens from the staged bilateral thoractomies. Because of the absence of neoplastic cells, immunohistochemical characterization of these potential cells was not warranted or possible. PRKACA fluorescence in situ hybridization (FISH) performed on a pulmonary nodule was inconclusive because of absent cellular tissue after the specimen underwent decalcification. No specimens had adequate tissue for FISH after decalcification. The operations were uneventful, and there were no postoperative complications.

Two weeks after the second stage thoracotomy, she continued being treated with 3 additional cycles of PLADO and sorafenib according to the AHEP 1531 group F (high-risk) protocol. New, tiny punctate bilateral calcified nodules were detected on end of therapy CT chest that will be closely followed on surveillance imaging. 

## 3. Discussion

Here, we report a case of diffuse nodular PO that mimicked pulmonary metastatic disease in a patient with fibrolamellar hepatocellular carcinoma. Because FL-HCC often presents with advanced disease, including pulmonary and lymph node metastases, and because this patient’s pulmonary nodules met PRETEXT radiographic criteria for metastatic disease, she was treated according to the high-risk HCC arm (unresectable or metastatic disease at diagnosis) on AHEP1531. Since the residual pulmonary nodules remained present and stable in size (although with increased calcification) after the conclusion of neoadjuvant chemotherapy, and complete excision of the primary hepatic tumor was obtained with negative margins, we recommended bilateral, staged thoracotomies to clear all evidence of potential disease. A right thoracotomy resulted in the excision of 11 total specimens, a greater number than was appreciated on preoperative CT scan. Nine of these lesions were consistent with nodular PO, and two were benign intraparenchymal pulmonary lymph nodes. Despite the pathology from the right-sided resections being compatible with a benign process, the patient and her family elected to proceed with the original plan for staged bilateral thoracotomies due to the remote possibility that these lesions represented differentiation of tumor cells given the transition from a non-calcified to a calcified appearance on CT scan during therapy. Therefore, a left thoracotomy was performed with resection of six nodules, five of which were consistent with nodular PO, and one represented a benign pulmonary lymph node. 

According to previous studies, nodular PO is usually found in patients with chronic elevated pulmonary venous pressure or left-sided heart failure [3,7]. However, this patient did not have a history of cardiopulmonary disease and had a normal echocardiogram prior to initiation of therapy. There are no reported associations between smoking marijuana and PO in the medical literature. The adjacent pulmonary parenchyma resected with each pulmonary nodule in this case did not demonstrate evidence of chronic lung disease or interstitial pulmonary fibrosis. The patient did not have a chest CT scan performed prior to her cancer diagnosis. Therefore, we were unable to compare with previous images, which may have suggested a chronic or benign process rather than FL-HCC metastases. These lesions were not apparent on plain chest radiograph. The pulmonary nodules’ size stability during induction chemotherapy could suggest a benign process since the primary tumor did slightly shrink during the initial treatment. However, according to RECIST criteria, the change in the primary tumor volume was consistent with stable disease. Therefore, the relative stability of the pulmonary lesions could be expected in FL-HCC, which is often refractory to chemotherapy treatment. The increased calcification of the pulmonary nodules during induction chemotherapy was worrisome for the osseous differentiation of tumor cells, as can be commonly found in the mesenchymal component of hepatoblastoma after neoadjuvant chemotherapy or has been reported in other tumor types, including osteosarcoma and carcinoid tumor [17]. Osseous metaplasia with bone marrow elements has also been documented in hepatocellular adenoma with malignant transformation to hepatocellular carcinoma and in primary hepatocellular carcinoma [18,19]. However, osseous differentiation of FL-HCC has not been described and there are no reports of ossesous differentiation of any metastastic HCC. One key difference between these previous reports of osseous metaplasia and the current case is that the metaplastic tumors lack the architectural organization observed in the current case and rather display disorganized, intermingled bone trabeculae and marrow elements.A careful pathologic examination of the resected pulmonary surgical specimens showed no evidence of a malignant process and no indication that these lesions represented such tumor differentiation. A representative nodule was sent for PRKACA FISH, however after decalcifcation of the lesion a hybridization signal could not be obtained due to lack of cellularity. Futhermore the nodules did not contain neoplastic cells that would be appropriate for immunohistochemical studies. We did not observe any evidence of bone metaplasia in the primary hepatic tumor in this case. Given the lack of malignant cells, the absence of bone metaplasia in the primary tumor, and the well-organized appearance of a circular arrangement of bone rimming the fat in the center of these nodules, we finally concluded that this condition more likely represented nodular pulmonary ossification than bone metaplasia. 

Patients with PO usually are asymptomatic, as was seen in the index case. Nevertheless, patients with diffuse pulmonary ossification (DPO), particularly the dendritic form, are more likely to be symptomatic. Two definitions of DPO were proposed by Egashira et al. [1]: 10 or more total bilateral nodular ossifications or one or more lobes with five or more ossifications. Our patient was found to have a left hepatic liver mass with multiple bilateral pulmonary nodules that exceeded ten nodules on final pathologic examination, and thus she meets the criteria for DPO. Twelve cases of the dendritic form of DPO complicated by spontaneous pneumothorax were reported. All the patients were male (46 +/− 17 years), and some of them had underlying chronic lung disease. Previous studies have also described middle-aged and elderly male patients with dendritic DPO who had exhibited slight declines in the diffusing capacity for carbon monoxide (DLCO) to 53.8%–64% predicted (normal ≥ 70%) [2,7]. Due to the lack of symptoms or evidence of a diffuse process, pulmonary function studies were not obtained in the current case but could be considered as the patient is followed-up long term. Pneumothorax and abnormal PFTs have not been described in patients with nodular PO without underlying lung disease. There are currently no standard treatments or follow-up guidelines for PO. 

The exact pathogenesis of this condition is still unclear, but it is hypothesized to be secondary metaplasia of pulmonary fibroblasts into osteoblasts in response to chronic inflammatory insults [1,4,5]. The dendritic type is more common in the setting of chronic pulmonary inflammation, including interstitial fibrosis [5,20]. Other conditions that have been associated with PO include amyloidosis, asbestos exposure, busulfan therapy, and cystic fibrosis. The current patient did not receive busulfan and had evidence of pulmonary nodules prior to the initiation of chemotherapy. Dendritic PO most commonly occurs in men during the fifth and sixth decades of life [5]. There was one study reporting two familial cases of DPO in father and son, and the author hypothesized that the dendritic form might be partly associated with familial genetic disorders and hypothetically associated with the bone morphogenetic protein (BMP) genes and the transforming growth factor-beta (TGF-b) [20]. Another study showed that dendritic DPO occurred in older men with chronic gastric acid aspiration [21]. One possible mechanism is that inflammation occurs within the lung interstitial space and replaces the interstitial structure with ossification. For the nodular subtype, it is hypothesized that this is the consequence of the biological response against alveolar exudate from pulmonary congestion [3,4,20]. 

On high-resolution CT, PO appears as 1–5 mm “ossified” intrapulmonary nodules in the lung periphery that are best appreciated on bone windows. PO often exhibits lower lobe predominance [1,21]. The differential diagnosis of small, calcified lesions on the CT scan includes pulmonary alveolar microlithiasis (but this is usually much smaller than PO, described as sand-like), small, calcified granulomas associated with granulomatous disease, and metastatic pulmonary calcification [1]. Calcification of metastatic nodules is uncommon and usually occurs with papillary thyroid carcinoma, adenocarcinoma, or carcinoid tumors. Osteosarcomas and chondrosarcomas after treatment may also become calcified [22].

We attempted to localize the pulmonary nodules by using NIR-guided ICG imaging. There are studies showing that NIR imaging could identify malignant pulmonary nodules as small as 0.2 cm and as deep as 1.3 cm from the pleural surface [16,23]. The failure to visualize any pulmonary lesions by ICG in our patient could be because all the resected lesions were benign. Pulmonary metastasectomy is indicated for patients with high-risk hepatoblastoma who demonstrate non-resolving pulmonary lesions during induction therapy. However, the role of pulmonary metastasectomy for FL-HCC is still equivocal because of the rarity of this disease. Nevertheless, it is well-established that patients with FL-HCC will not survive without achieving no evidence of disease during treatment. Therefore, an aggressive approach to metastatic lesions is currently advised if all gross disease can be feasibly removed [24,25,26].

If the pulmonary lesions had been definitively known to be PO at diagnosis, the patient would have most likely undergone up-front left hepatectomy and been assigned to AHEP1531 Group E2—Resected at Diagnosis—de novo HCC. She would have then received four cycles of adjuvant PLADO. However, definitive diagnosis of the pulmonary lesions by biopsy at diagnosis is not recommended in the AHEP1531 protocol and likely would have required thoracotomy. The post-biopsy pathology would have been controversial and this could have resulted in delayed cancer therapy. 

## 4. Conclusions

In conclusion, diffuse PO is a rare pulmonary condition that could mimic pulmonary metastatic disease if encountered during cancer therapy and should be included in the differential of calcified pulmonary lesions identified in cancer patients.

## Figures and Tables

**Figure 1 children-08-00226-f001:**
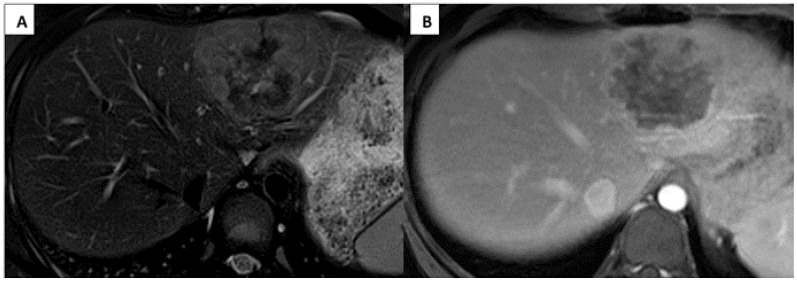
Imaging appearance of fibrolamellar hepatocellular carcinoma. Axial MRI liver demonstrates the PRETEXT II single left hepatic mass in segments II, III, and IVa with mild peripheral hyperintensity and central hypointensity on T2 fat saturation images (**A**) and iso to hypointensity on the post contrast phase (**B**).

**Figure 2 children-08-00226-f002:**
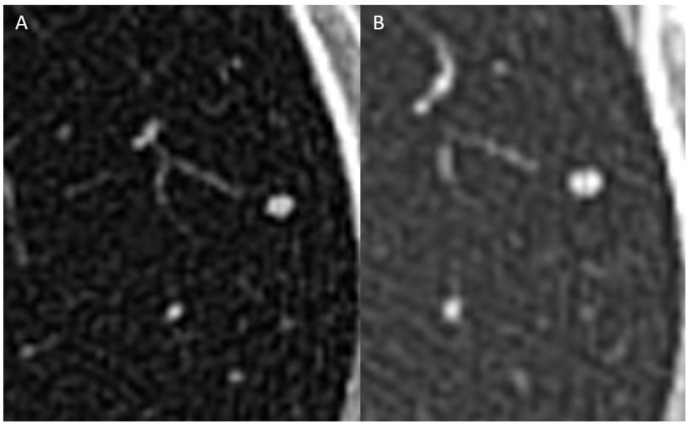
Magnified images from 0.625 mm slice non-contrast Computed Tomography (CT) scan at baseline (**A**) and contrast-enhanced CT 2 months later (**B**) show a 3 mm pulmonary nodule in the left upper lobe with possible calcification on the later study.

**Figure 3 children-08-00226-f003:**
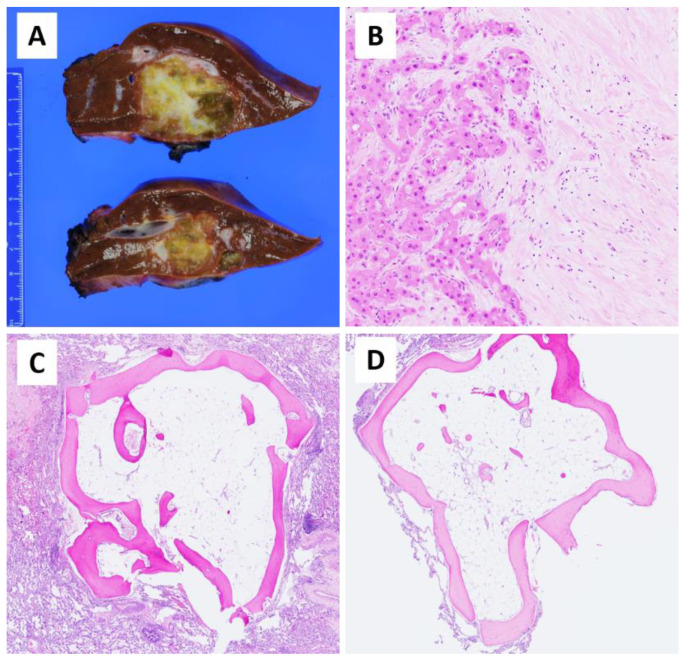
Pathological findings of a patient with Fibrolamellar Hepatocellular carcinoma (FL-HCC) and bilateral diffuse nodular pulmonary ossification (PO). Cross-section of the left hepatectomy showing a 5.6 × 5.0 × 3.6 cm tan-gray, necrotic appearing, and relatively well-circumscribed mass (**A**). Post-chemotherapy FL-HCC with areas of viable residual tumor (left side) composed of thick trabeculae of large polygonal cells separated by dense desmoplastic collagenous stroma aside areas of necrosis and chemotherapy-induced changes (right side). Hematoxylin & Eosin, 200× (**B**). Lung parenchyma with multiple nodules composed of a rim of well-organized lamellar bone and mature adipose tissue consistent with diffuse nodular PO. No inflammatory infiltrate, no granulomatous reaction, and no tumor cells were noted. Hematoxylin & Eosin, 40× (**C**,**D**).

## Data Availability

This study does not include reporting of any data that would warrant being deposited in a publicly archived database.

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
