# Peer review of "Bilateral Diffuse Nodular Pulmonary Ossification Mimicking Metastatic Disease in a Patient with Fibrolamellar Hepatocellular Carcinoma"

_children, 2021, doi:10.3390/children8030226_

Round 1
Reviewer 1 Report
The manuscript is based on a single case report and is well written. However the diagnosis which was the basis of the case report was not well-documented:
- Bone metaplasia has been well-documented in carcinomas and liver tumors in general. Few cases have been reported of hepatocellular carcinoma with osseous metaplasia: Clin Gastroenterol Hepatol. 2015 Mar;13(3):e26-7. and J Pathol Transl Med. 2018 Jul;52(4):226-231. Hence, their discussion point in lines 169-171 is not valid.
- The lung specimens have been resected and examined after a course of chemotherapy which could have altered the tumor morphology
- The authors failed to document absence of tumor in the lung specimens with bone. Detailed images, immunohistochemical tests and NGS molecular testing documenting absence of DNAJB1-PRKACA fusion should have been performed on the lung specimen.
- Lack of etiologic factors associated with pulmonary ossification such as chronic inflammation and portal hypertension, makes such diagnosis unlikely.
- The argument that this could represent a first case report of pulmonary ossification associated with fibrolamellar hepatocellular carcinoma is less plausible than a case of hepatocellular carcinoma with osseous metaplasia. Both conceptions have not been reported before.
Author Response
Reviewer 1:
The manuscript is based on a single case report and is well written. However the diagnosis which was the basis of the case report was not well-documented:
- Bone metaplasia has been well-documented in carcinomas and liver tumors in general. Few cases have been reported of hepatocellular carcinoma with osseous metaplasia: Clin Gastroenterol Hepatol. 2015 Mar;13(3):e26-7. and J Pathol Transl Med. 2018 Jul;52(4):226-231. Hence, their discussion point in lines 169-171 is not valid.
Response: We acknowledge in the manuscript that bone metaplasia is on the differential. We have added the suggested references that document bone metaplasia in cases of hepatocellular carcinoma. However, because of the absence of bone metaplasia in the primary tumor in the current case, the well-organized appearance of the nodules compared to the typical disorganized architecture of bone metaplasia, and no neoplastic cells or treatment effect observed in the pulmonary resection specimens, we feel that this condition more likely represents nodular pulmonary ossification than bone metaplasia. We have revised and added to the section of the discussion noted by the reviewer (lines 176-192).
- The lung specimens have been resected and examined after a course of chemotherapy which could have altered the tumor morphology
Response: This is correct and this possibility is acknowledged and specifically stated in lines 172-175 in the discussion section. This possibility also is what prompted the family to continue along with the original plan for bilateral thoracotomies as mentioned in the manuscript.
- The authors failed to document absence of tumor in the lung specimens with bone. Detailed images, immunohistochemical tests and NGS molecular testing documenting absence of DNAJB1-PRKACA fusion should have been performed on the lung specimen.
Response: The manuscript has been updated to confirm that no neoplastic cells were detected in any of the 17 pulmonary resection specimens from the staged bilateral thoracotomies (Lines 121-122). Therefore, immunohistochemical characterization of neoplastic cells was not possible. PRKACA FISH was performed on a representative nodule, but after the requisite decalcification process for these highly calcified nodules, insufficient cellularity was present to enable an appropriate hybridization signal. This information has been included in the revised version of the manuscript (lines 121-126.
- Lack of etiologic factors associated with pulmonary ossification such as chronic inflammation and portal hypertension, makes such diagnosis unlikely.
Response: The association between nodular pulmonary ossification and chronic pulmonary inflammation is included and discussed in the manuscript (lines 160-165). The only potential source of chronic pulmonary inflammation in the current case is recreational cannabis smoking. This is briefly mentioned in the case report; however, we did not want to extensively discuss this detail because it would be only speculation.
- The argument that this could represent a first case report of pulmonary ossification associated with fibrolamellar hepatocellular carcinoma is less plausible than a case of hepatocellular carcinoma with osseous metaplasia. Both conceptions have not been reported before.
Response: We believe that osseous metaplasia is less likely than diffuse pulmonary ossification because no neoplastic cells were detected in any of the 17 pulmonary resection specimens, the pattern of histology is much more organized (circular arrangement of bone encircling fat cells) than bone metaplasia, and the primary tumor in this case did not exhibit any osseous metaplasia. However, osseous metaplasia is discussed as a possibility on the differential and that influenced our clinical decision-making in the manuscript. Therefore, we feel that we have adequately considered and discussed this possibility.
Reviewer 2 Report
Very well written, novel contribution to literature.
Appropriate details related to diagnoses and differentials and thorough description of patient's evaluation and course.
It might be interesting to outline if/how earlier identification of PO as opposed metastases might have changed management but understand if authors feel that this is beyond scope. Authors appropriately point out that patient had all evaluations as recommended per current protocol and met protocol definition of metastatic disease and group F treatment.
Author Response
Reviewer 2:
Very well written, novel contribution to literature.
Response: Thank you for your supportive comments.
Reviewer 3 Report
This was a well-written case report documenting a rare but important finding. Excellent discussion on fibrolamellar HCC as well.
Author Response
Reviewer 3:
Appropriate details related to diagnoses and differentials and thorough description of patient's evaluation and course.
It might be interesting to outline if/how earlier identification of PO as opposed metastases might have changed management but understand if authors feel that this is beyond scope. Authors appropriately point out that patient had all evaluations as recommended per current protocol and met protocol definition of metastatic disease and group F treatment.
Response: This is an important comment. The reality is that earlier or definitive identification of PO may have significantly altered this patient’s risk stratification. The characterization of the pulmonary lesions as metastatic disease resulted in high-risk stratification and bilateral staged thoracotomies. This is mentioned in Lines 49-50 of the manuscript. In lines 241-246 of the revised manuscript, we now address how risk stratification and treatment would be different had a definitive finding of PO been possible at diagnosis.
This was a well-written case report documenting a rare but important finding. Excellent discussion on fibrolamellar HCC as well.
Response: Thank you for your supportive comments.
Round 2
Reviewer 1 Report
The authors have satisfactorily addressed all my concerns